# Fast and Reliable Sending of Generic Object Oriented Substation Event Frames between Remote Locations over Loss-Prone Networks

**DOI:** 10.3390/s23218879

**Published:** 2023-11-01

**Authors:** Jose Saldana, Aníbal Antonio Prada Hurtado, Eduardo Martinez Carrasco, Yasmina Galve, Jesús Torres

**Affiliations:** CIRCE Technology Center, Avenida Ranillas, 50018 Zaragoza, Spain; aaprada@fcirce.es (A.A.P.H.); emartinez@fcirce.es (E.M.C.); ygalve@fcirce.es (Y.G.); jtorres@fcirce.es (J.T.)

**Keywords:** smart grid communications, network impairments, packet loss, latency, WAMPAC, substation protection, IEC 61850, GOOSE, telecommunications for POWER systems, VPN

## Abstract

WAMPAC (Wide Area Monitoring Protection and Control) applications are becoming crucial for granting a stable operation of the electricity transmission grid. These systems use a set of sensors distributed between different electrical substations to gather real-time measurements from the field. These sensors are called Phasor Measurement Units (PMUs). Using the gathered data, different monitoring, protection, and control algorithms are run in a Phasor Data Concentrator (PDC) located in a central location. These algorithms close the loop via the generation of remedial commands, which are sent back to the field level with stringent delay, security, and reliability requirements. GOOSE (Generic Object Oriented Substation Events) protocol, defined by IEC 61850 (IEC stands for International Electrotechnical Commission), is used for that aim and also considers the option of sending these commands over IP networks (this option is called Routed-GOOSE). The present article proposes two alternatives for the tunneling of GOOSE frames over IP. Both options allow the decoupling of the transmission and the security aspects, thus increasing flexibility and allowing for easier deployment. The first option, called VX-GOOSE, is a combination of standard protocols, allowing the sending of these frames over UDP/IP tunnels. The tests that have been carried out demonstrate that, under certain network conditions, the transmission of GOOSE frames over UDP may fail, and in some extreme cases, even a whole burst of GOOSEs could be lost. This may have very bad consequences for a distributed electrical system. It should be noted that this limitation affects both VX-GOOSE and Routed-GOOSE. To overcome these limitations, the second option, called Simplemux *blast* mode, includes a novel mechanism that provides delivery guarantees and a reduced delay, with the counterpart of a certain degree of redundancy. As shown in the experiments, the incurred delays can be significantly reduced when remote locations are connected via unreliable networks, whereas the bandwidth increase caused by redundancy can be kept at reasonable levels. Finally, it should be remarked that although GOOSE is a relevant example use case, this approach can be applied in other fields where flows require very low delay and delivery guarantees.

## 1. Introduction

Cross-border electricity interconnections are necessary to establish a geographically large market in which major stakeholders of the energy value chain can cooperate. These markets, based on imports and exports of electricity, increase the level of competition, enhance the security of supply, and permit a better integration of renewable energy sources. In this context, the use of WAMPAC (Wide Area Monitoring Protection and Control) [1] systems is crucial to grant a stable and seamless integration of the grids of different countries. A WAMPAC integrates different elements: first, a set of sensors called PMUs (Phasor Measurement Units) are distributed in different electrical substations throughout the system. They send their real-time measurements of electrical quantities (usually called synchrophasors) to a remote central controller called PDC (Phasor Data Concentrator), where they feed monitoring, protection, and control algorithms, able to detect any impairment or instability. Depending on the output of these algorithms, remedial actions or actuation orders can be issued back to field-level devices. This “closes the loop” of the system, reacting to potential or detected problems in a fast way.

Cross-border WAMPAC systems [2] present some specific challenges that must be addressed properly. First, the geographical distance between the sensors, the central controller, and the actuators produces inevitable delays in the order of some tens, or even cents, of milliseconds. This must be handled properly to ensure that the algorithm actuation times comply with the applicable regulations. In addition, since digital measurements are involved, precise synchronization is required between all the elements.

Many utilities are nowadays connected via dedicated networks, but the trend toward a fully IP smart grid is gaining more traction in terms of cost and bandwidth [3]. In some use cases, although it would be desirable to avoid the use of IP networks, this may prove unavoidable. Consequently, the protection or control equipment is linked to extensive communication networks, the performance of which cannot be fully controlled or known. This corresponds to the use cases defined in IEC 61850-90-5 [4] (IEC stands for International Electrotechnical Commission), stating that IP networks can be used to communicate with receivers outside a substation if the added delays are acceptable for the application. The sending of Ethernet frames over other technologies is defined in IEC/TR 61850-90-1 [5].

Some substations that are not connected to a dedicated network may use a 4G wireless one. This happens, for example, in a WAMPAC system under development in the context of the H2020 FARCROSS project [2], where the interconnected electrical grids of different countries have been used to test these tools.

The use of a network with a more random behavior (e.g., a wireless one) instead of a dedicated one raises two kinds of concerns: first, cybersecurity is a must in these scenarios, considering the primary importance of continuous electric service. The use of Virtual Private Networks (VPNs) between remote locations can provide a high degree of security. However, substation automation standards define their own security mechanisms, which may require additional implementation and resource effort. Second, the variability of the network parameters (delay, jitter, packet loss, and bandwidth limits) is much higher than that of a dedicated one, and these network impairments will present a more severe profile.

In this context, the contribution of the present paper is focused on exploring two suitable solutions for using tunnels to send event-driven field commands that can fulfill the presented needs in WAMPAC systems. The approach can be summarized as follows:First, the proposal of a novel combination of standard protocols that decouples the security from the transmission of information. It is called VX-GOOSE and uses VXLAN (Virtual Extensible LAN) [6] to send tunneled GOOSE (Generic Object Oriented Substation Event) frames (the ones carrying the event-driven commands), allowing the transmission of actuation events between equipment from different vendors via IP networks in a fast and secure way.Second, the proposal of a mechanism called Simplemux *blast* flavor, based on sending redundant frames, which grants the delivery of every single frame and minimizes the delay caused by packet loss, thus keeping actuation times within acceptable limits. This proposal has been designed after discarding the use of TCP and SCTP (Stream Control Transmission Protocol) as suitable options for this kind of traffic.And third, a set of tests with real hardware, demonstrating and comparing both proposals and measuring the effects of network impairments on their performance.

The added value of this work is that both proposals serve as implementations of the tunneling mechanism for Ethernet frames over IP, proposed by IEC 61850-90-1. In addition, the latter offers another feature: it periodically re-transmits GOOSE messages until an acknowledgment is received, thereby reducing delay in loss-prone networks. This can be an added value in scenarios where these kinds of networks must be used. In addition, these options can be deployed in a fast and simple way. The paper offers an assessment of both proposals using real-world implementations, a crucial step before considering actual deployment.

To prevent unauthorized access, the two proposed approaches should be used in conjunction with one of the robust existing VPN solutions. Since security is separated from transmission, the responsibility of cybersecurity aspects can be lifted from the grid control staff and managed by a specialized team.

The remainder of the paper is organized as follows: the next section summarizes the state of the art regarding protocols for substation automation, also considering security and latency issues; Section 3 explains VX-GOOSE and presents the corresponding evaluation tests; Section 4 does the same with Simplemux, *blast* flavor, and the paper ends with the Conclusions.

## 2. Related Work

This section is organized into three subsections, each addressing a distinct topic relevant to current research. To facilitate comprehension, a summary table of the related work is provided at the conclusion of each subsection.

### 2.1. Protocols for Substation Automation

IEEE C37.118 [7] is the dominant protocol for the transmission of synchrophasor data [8], i.e., the measurements of the electrical quantities. This standard defines a protocol for the transmission of synchronized phasor measurement data among power system devices. It details the types, usage, content, and data formats of messages for real-time communication, specifically between Phasor Measurement Units (PMUs), Phasor Data Concentrators (PDCs), and other elements.

The second part of C37.118 defines the data transmission format (see Figure 1). It can travel on TCP or UDP datagrams over IP. The other key standard for measurement digitalization is *Sampled Values* (SV), defined in IEC 61850-9-2 [9] and adopted in IEC 61869-9 [10]. It was initially conceived to operate within a substation, going directly over the 802.3 protocol (Ethernet).

In addition to the transmission of samples via SV, IEC 61850-8-1 [11] defines a protocol for event notification (GOOSE). It is designed for quick, widespread notification and related execution of events, commands, or trips within the substation via the IED (Intelligent Electronic Device) in charge of operating a given substation switchgear element. As an example, a *trip* may result in the opening of a switch when an electrical fault has been detected. In some cases, these *trip* orders can be very critical, and their loss or delay can make the difference between a blackout and a simple outage.

GOOSE protocol travels over Ethernet, so the publisher does not receive any confirmation from the subscriber, and there is no retransmission method. To minimize the chance of message loss, each GOOSE frame is sent a number of times (in a burst). The interval between frames is increased at each transmission until a steady inter-frame time is reached.

IEC 61850-90-1 [5] gives an overview of all the aspects to be considered when IEC 61850 is used for the exchange of information between different substations. It proposes two mechanisms of communication between different Local Area Networks (LANs) [12]: *Tunneling* and *Gateway*. The former can be defined as any mechanism that passes a message through a network without any modification. The document does not specify the kind of tunnel to be used or any particular implementation. In this context, the present work proposes a practical approach to that solution facing the potential problems and the benefits to be obtained from the tunneling solutions. Other works considering tunneled GOOSE messages over IP networks are [13], where a set of tests with simulated GOOSE traffic was run over a GRE (Generic Routing Encapsulation, [14]) tunnel; and [15], where L2TPv3 (Layer Two Tunneling Protocol—Version 3 [16]) was used to send GOOSE frames via mobile networks for a Logic Selectivity application.

Later, IEC 61850-90-5 [4] defined a *Routed* version of SV (known as *R-Sampled Values*) and GOOSE (known as *R*-GOOSE) over IP networks, which may also be used for WAMPAC systems. Both *R*-SV and *R*-GOOSE can travel over UDP via IP networks [17]. Some studies [8] prove that GOOSE has been widely deployed within substations. However, *R*-GOOSE has not reached the same spread, and very few market devices include it in their configuration.

In contrast with *Sampled Values*, C37.118 does not have an equivalent “companion protocol” for the transmission of events, so some implementors have developed their own proprietary extensions, leveraging on C37.118 *extended command* frames. These custom commands lead to vendor-specific solutions, hampering interoperability.

To put our two proposals in context, both of them (VX-GOOSE and Simplemux *blast* flavor) can be considered as a way to implement the *Tunneling* mechanism proposed by IEC 61850-90-1 over IP. The latter, being also a *Tunneling* mechanism, includes an additional feature, i.e., the periodic retransmission of GOOSE messages (until an acknowledgement arrives) to reduce the delay in loss-prone networks.

Table 1 summarizes the related work, including standard definitions, reports, and research papers.

### 2.2. Security

Substations are critical infrastructures, so cybersecurity is a crucial aspect when designing their communication schemes. The IEC/TS 62351 series [18] is designed to secure the TC 57 series of protocols. Its security objectives include a range of measures: authenticating data transfers via digital signatures, ensuring access is only granted to authenticated users, preventing eavesdropping and spoofing, blocking playback, and detecting intrusions. Both *R*-GOOSE and *R*-*Sampled Values* can be encrypted and authenticated according to the recommendations of the IEC 62351 standard.

As the security model for IEC 61850-90-5 [4] is based upon the threats and functions found in the IEC/TS 62351 series [18], this makes it necessary for vendors to jointly implement the security and the synchrophasor transmission protocols, also considering other requirements such as the added delay [19]. Since both issues are tightly coupled, the potential limits or problems of one of them would affect both.

In contrast, C37.118 [7] does not define any native security protocol, so production implementations usually opt to deploy end-to-end VPNs [20]. As security is decoupled from transmission, staff in charge of grid control are relieved from cybersecurity aspects, which can be managed by a specialized team.

Different VPN options with a high level of security can be considered for this aim. We will summarize the characteristics of some of the most popular ones: WireGuard [21] is natively supported in Linux and other operating systems. Its traffic travels over UDP with no delivery/ordering guarantees. This particular feature is especially interesting in our case, as it avoids any extra delay caused by packet reordering. OpenVPN [22] is another popular option. It is a user-space application that relies on OpenSSL and enables TLS (Transport Layer Security) support. IPsec (Internet Protocol security [23]) is a suite including two protocols (Encapsulated Security Payload and Authentication Header) and two modes (Tunnel and Transport). It is a widely used IETF (Internet Engineering Task Force) standard.

In [21] and also in [24], four of the most popular VPN solutions were benchmarked, namely WireGuard, IPsec (in different modes), and OpenVPN. The results showed that OpenVPN has the lowest throughput and the highest latency; being a user space application, it incurs some delays when copying packets between user space and kernel. IPsec with AES-based (Advanced Encryption Standard) encryption and WireGuard presented a similar performance. While IPsec performed better in virtualized environments, WireGuard outperformed IPsec suites in non-virtualized ones due to its simple implementation and low overhead. Finally, some other options, such as GRE [14], were also compared in [25].

Regarding the overhead added by the VPN, this is not a problem in our use case: the size of the GOOSE frames is typically 150–300 bytes), and the VPN may require between 40 and 60 extra bytes, so the total size is very far from the Maximum Transmission Unit size (1500 bytes for Ethernet).

Table 2 summarizes the related work, including standard definitions and related research papers.

### 2.3. Latency

In [26], the different components of network delay were classified, and the concept of “latency budget” was defined, which is “consumed” by different sources of delay. These components can be associated with the typical delays found in the data networks of electrical systems:*Generation*: the time between a physical event and the availability of data. In our case, this would correspond to the *Fault Recognition Time* (the time to detect the fault) plus the *Time for Initiating Transmit Action* defined in the CIGRE (International Council on Large Electric Systems) report *Protection using Telecommunications* [27].*Transmission*: this would be the *Propagation Time* defined in [27] plus the time required for generating the packet, i.e., the one that depends on the packet size. In a wide area network, the *Propagation Time* is usually in the order of the tens or hundreds of milliseconds [28], whereas the generation time is smaller: as an example, in a 100 Mbps network, 100 bytes are sent in 8 μs. Considering that GOOSE frames may be between 150 and 300 bytes, this time can be considered negligible.*Processing*, *Multiplexing*, and *Group/batching*: all the elements add their respective delays. The *Selection and decision time* and the *Additional delay due to disturbance* defined in [27] will be included here.

IEC 61850-5 establishes a limit of 3 ms for GOOSE frames for *Type 1A Trip* traffic (a kind of traffic that does not leave the local network of a substation). This delay limit is defined for local area networks (LANs), i.e., the ones that are deployed in substations. However, for a WAMPAC system, the propagation time via large communication networks must be considered. This latency can be significant: the delay in an optical fiber is roughly 5 µs/km, whereas in a radio link, it is 3.3 µs/km.

As stated in [27], the propagation time across the network is a critical parameter, and it has to be kept to a minimum so as to ensure a fast trip of a circuit. The use case considered in the present paper would correspond to the one called “*Teleprotection connected* via *telecommunication network*” in that document.

Empirical measurements of intra-continent connections [28] draw values of round-trip time of about 15 ms (Europe, Japan), 30 ms (North America), 60 ms (Latin America), or 85 ms (Asia Pacific). The one-way delay is estimated to be approximately half of the round-trip time.

According to [27], the fault-clearing time for a protection system should be between 42 and 210 ms. The *Teleprotection operating time*, which is a part of it, should be between 2 and 70 ms. In our case, this would correspond to the latency budget: in some scenarios, the transmission delay would be its main component, so the rest of the delays must be kept as low as possible to grant a good performance of the protection functions of the WAMPAC. Nevertheless, it should be noted that the geographical distance entails a delay that is unavoidable, and this fact must be kept in mind when designing a WAMPAC.

Table 3 summarizes the cited work.

## 3. VX-GOOSE

In order to send WAMPAC’s remedial actions, it would be convenient to have a widely deployed protocol that can travel through IP networks. However, none of the existing options seems easy to implement. GOOSE is widely deployed, but it is restricted to the LAN level because it travels directly over Ethernet; *R*-GOOSE, although conceived for traveling over IP, is not yet very popular; finally, proprietary extensions of C37.118 hamper interoperability.

### 3.1. Description of VX-GOOSE

As a first solution, this paper proposes VX-GOOSE, which consists of using a standard called VXLAN [6] in combination with GOOSE (the right column of Figure 1, in which VXLAN would be the *tunneling* protocol). VXLAN is a protocol for network virtualization over Layer 3, defined by the IETF, originally created to overcome the limitation in the number of VLANs (Virtual Local Area Networks) in data centers. It adopts the MAC-in-UDP packet encapsulation mode, also including a specific 8-byte header with an identifier. This way, the entire network (including switches at different locations) becomes a large Layer-2 virtual switch.

On behalf of clarity, a Wireshark capture of a GOOSE frame traveling over VXLAN over UDP (port 4789) is shown in Figure 2.

The advantages of VXLAN with respect to other tunneling protocols [13,15,16] are its flexibility and speed (no tunnel nor session setup phases are required) and its high scalability, as it was initially conceived for data center hosting thousands of machines in different LANs, the number of locations it can connect is huge.

As illustrated in Figure 3, what is proposed is a new use case for VXLAN. GOOSE travels over Ethernet frames, which are captured at the origin switch (e.g., at the control center) and sent through a tunnel via a WAN (Wide Area Network) IP network (i.e., to the remote substation). This way, the whole Ethernet frame “appears” in the destination switch, making the end device “think” that it has been originated locally. The same happens backward: bidirectional GOOSE can work normally since the tunnel is transparent for both communication ends. In the figure, the RTAC is a Real-Time Automation Controller, i.e., the machine where the protection algorithms run.

Section 11.3.1.3 of IEC 61850-90-5 defines the differences between GOOSE and *R*-GOOSE data: it recommends that the *DataSet* elements include a timestamp. It also suggests that the *QUALITY* for each *DataSet* element may need to be included. None of these changes is in any way critical for this proposal.

As can be seen, the use of a VXLAN tunnel combines two advantages: First, it has the benefits of *R*-GOOSE, as all the functionality of GOOSE is maintained, but without the need to implement all the specific features of *R*-GOOSE. Second, the tunnel can make use of a VPN (which may already exist to secure the connection between remote locations) so security can be decoupled from the transmission of information.

### 3.2. Tests with VX-GOOSE

A setup with real equipment has been used to validate the proposal (Figure 4): a Real-Time Digital Simulator (RTDS) simulates an electric grid [29]. A PMU sensor (SEL Axion 2240) obtains the measurements from the simulated grid and sends C37.118 synchrophasors to a Real-Time Automation Controller (RTAC, SEL 3555), where protection algorithms are run [30]. Using VXLAN, the local (SEL 2730M) and the remote (TP-Link SG108E) switches are connected as if they were in the same LAN. The VXLAN routers are two Raspberry Pi 3B+ (Linux kernel 5.10.17) that capture the traffic and send it through the tunnel to the other end. This way, the very same frame generated by the protection algorithm in one location is released at the destination switch, and it arrives at the destination PMU, where an actuator is triggered.

In this setup, two network impairments must be considered to emulate the network behavior: (a) delay: its effect is direct, i.e., it affects VX-GOOSE packets, adding latency to the execution of the algorithm decision; and (b) packet loss: VX-GOOSE travels on UDP, so there are no retransmissions. GOOSE mitigates this by sending a burst of packets with the same content. If a packet is lost, some of the subsequent ones may arrive, so a lost packet is translated into an additional delay on the protection algorithm. An interesting research question arises: considering the bursty nature of packet loss on IP networks [31], can this represent a problem for *R*- and VX-GOOSE?

A battery of tests was run to answer the question. A wide area protection algorithm is running in the testbed, based on the Zone Integrated Impedance Angle method [32], applied to a 400 kV transmission line. Each test consists of 40 faults, forced by the RTDS every 22 s. A burst of GOOSE frames is generated by the RTAC after each fault. Random packet losses are introduced in the network using Linux *netem* with a Simple Gilbert Model [33], which provides a good approximation of losses on the Internet. It has two parameters, *p* and *r*, corresponding to the transition probabilities between the *bad* (all packets are lost) and the *good* (no packet loss) states (see Figure 5).

The two Raspberries are synchronized via NTP (Network Time Protocol) before the test. Wireshark is used on both sides to capture all the traffic. Once the test is finished, both capture files are cleaned and parsed using a Python script. Then, they are compared in order to obtain the delay of each packet and to identify the ones that have been lost.

On behalf of clarity, Table 4 includes the employed variables and their definitions.

We will use the next two parameters for the graphical representation of the results: first, packet loss probability, obtained as the probability of being in a *bad* status:(1)$$P_{loss}=\frac{p}{p+r}.$$

In addition, the Average Burst Error Length (ABEL), which is calculated as
(2)$$\mathrm{ABEL}=\frac{1}{r}.$$

Table 5 presents the average results, and Figure 6 shows a test with 10% packet loss and ABEL = 5 packets (a very harsh test setup where the bad effects of the network can be clearly observed).

During normal operation, GOOSE messages are generated every second (this corresponds to the *normal* GOOSE frames of the figure). After a fault, a burst of GOOSE frames is generated, which interval is increased at each transmission until the steady periodicity value is reached.

If the first frame of a GOOSE burst is lost, the message will arrive with an additional delay (the time interval between the first and the second GOOSE). If the second frame is also lost, the delay will increase, and so on. Furthermore, if the packet loss probability is high, a whole burst of GOOSE packets can be lost, as has happened with test #35, presented in Figure 6 (highlighted in yellow).

As can be observed from the table, if packet loss does not happen in bursts (ABEL = 1), only minimal delays appear. Even with a 10% loss rate, the average delay is only 0.6 ms. However, if long bursts of lost packets happen (ABEL = 5 or 10), the delays increase up to 24.9 ms. Furthermore, the combination of a high loss rate (about 10%) with long error bursts (ABEL = 5 or 10) can even cause a trip failure.

All in all, the tests have demonstrated that, under certain network conditions, the transmission of GOOSE frames over UDP may fail, and in some cases, even a whole burst of GOOSEs could be lost. This may have very bad consequences for a distributed electrical system. It should be noted that this statement is valid for both VX-GOOSE and *R*-GOOSE (the standard proposed by the IEC). Therefore, new mechanisms are needed to eliminate this possibility.

## 4. Simplemux, *Blast* Flavor

For an electric network operator, it is desirable to have a dedicated connection between the control center and each of the remote locations (substations). However, this is not yet the case in many scenarios where the operators resort to public IP networks or other solutions. Although nowadays’ wireless networks may provide good performance and a high throughput, their loss rate is still not negligible (0.1 to 0.5%). In addition, the bursty nature of packet loss may result in the loss of all the GOOSE frames of a trip, as has been observed in the previous section. This is something that should never happen in a real network since it would prevent a protection algorithm from acting.

### 4.1. Possibility of Tunneling over TCP or SCTP

A possibility that could be considered to totally avoid packet loss would be to send the GOOSE frames over TCP, a protocol that provides delivery guarantees. However, TCP retransmissions require at least an extra exchange of packets between the sender and the receiver, i.e., a latency equivalent to the RTT (Round-Trip Time), in addition to the timeout expiration.

To test the suitability of using a TCP tunnel, we have resorted to Simplemux [34], a protocol able to encapsulate a number of packets/frames belonging to different protocols into a single IP packet. In *normal* flavor, it just adds a small separator before each of the aggregated packets/frames. The encapsulated packets/frames can travel over IP and UDP.

In the present work, the possibility of traveling over TCP has been added to an existing user space implementation of Simplemux. The implementation is available at https://github.com/simplemux/simplemux (accessed on 30 October 2023). A Wireshark screenshot is shown in Figure 7, obtained with a *.lua* Simplemux dissector added as a plugin. It can be observed how Simplemux allows the sending of GOOSE frames over TCP packets using port 55557. A GOOSE frame with a size of 242 bytes is now sent inside a 311-byte frame. Ethernet, IP, and TCP add an overhead of 14, 20, and 32 bytes each (the TCP header has some extensions in this case), while Simplemux adds 3 more bytes. It can also be observed that the Simplemux header includes the length and the protocol code 143, which corresponds to Ethernet.

After some testing in the lab using Simplemux to send GOOSE frames over TCP, it was observed (see Figure 8) that in some cases, the delay incurred was up to 220 or even 455 ms (this happened with a 1% loss rate, ABEL = 1, RTT = 5 ms). In the figure, it can be observed that four trip bursts were seriously affected by these delays (bursts #3, #10, #27, and #35, highlighted in yellow).

Furthermore, if the loss conditions become harder, especially if ABEL is higher, TCP stops working; it disconnects and needs a long reconnection time. Obviously, this is not an acceptable solution in our case since a remote command must be executed in a fast way: if a fault has been detected in the grid, the time to act is critical.

An alternative to UDP and TCP is SCTP, which is also a widely accepted standard with many mature implementations; it was published in 2007 [35], and was updated recently [36]. It has a congestion control mechanism similar to that of TCP (including features such as Slow Start, Congestion Avoidance, Fast Retransmit, and Fast Recovery [37]). Therefore, the same limitations observed with TCP will apply.

All in all, it can be said that although the retransmission features of both TCP and SCTP make them able to grant that every single packet is delivered, they may add some delays that can be too high for this specific use case. In addition, their congestion control mechanisms may reduce the throughput [38], and this is not the desired behavior, considering that certain equipment may be at risk.

### 4.2. Description of Simplemux, Blast Flavor

Once the use of tunneled GOOSE over TCP or SCTP has been discarded, new options have to be proposed. An interesting fact is that the throughput of a GOOSE flow is quite minimal: some tens of kilobits per second. Therefore, a possibility is to add a certain degree of redundancy, repeatedly sending each frame a number of times until it is acknowledged by the other side.

For that aim, a new flavor, called *blast*, has been designed and added to the existing Simplemux implementation. It redundantly sends the same packet a number of times. Its protocol stack corresponds to the one in the right column of Figure 1, in which Simplemux would be the *Tunneling* protocol. For clarity, a Wireshark capture of Simplemux, *blast* flavor, travelling over UDP port 55558 is shown in Figure 9. In this case, the frame is 277 bytes long; the original GOOSE had 229 bytes, plus 14 bytes of the Ethernet header, 20 of IP, and 8 of UDP. Finally, the Simplemux header adds 6 more bytes. More details about the protocol fields and their values are given in Appendix A.

As shown in Figure 10, a period is defined: each frame sent by the RTAC is stored in the sender router and sent periodically via the tunnel until the first acknowledgment arrives. For that aim, application-level ACKs (Acknowledgements) are used. This increases the required throughput, but it guarantees that every single frame will arrive on the other side. Then, the destination router decapsulates the received frame and forward it to the end node.

It should be noted that since the mechanism works between a pair of intermediate machines, it is totally transparent for the end nodes, which only receive a single copy of the original frame. This is quite different from TCP: the proposed method does not wait for the ACK; it periodically sends a copy of the same frame to the other side. In high RTT networks, this can significantly reduce the incurred delay: instead of waiting for the whole RTT, a copy of any lost frame will soon be available.

As can be observed in Figure 10 (frame #1), if a tunneled frame is lost, a new copy will be available after an interval similar to the defined period. If a number of packets *l* are lost at the beginning of a burst, the additional delay becomes period×l (see frame #2). However, if the lost packet is not the first copy (see frame #3), the loss is not relevant.

To make an analysis of the incurred throughput increase, a parameter called *redundancy factor* (*R*) can be defined as
(3)$$R=\frac{number\;of\;tunneled\;frames\;sent}{number\;of\;original\;frames}=\frac{RTT}{period}+E\left[l\right].$$

If a number of packets *l* is lost at the beginning of a burst, this will be translated into an additional delay:*Additional delay* = period × E[*l*].(4)


To obtain E[*l*], let *P_loss_* be the loss rate. Let *k* be the number of packets in a row that are lost. The number of tunneled frames lost at the beginning of a burst will be
(5)$$E\left[l\right]=\left(1-P_{loss}\right)\overset{\infty }{\underset{k=0}{\sum }}kP_{loss}{}^{k}.$$

The closed form of the sum is
(6)$$\overset{n}{\underset{k=0}{\sum }}kp^{k}=\frac{1-\left(n+1\right)p^{n}+np^{n+1}}{{\left(1-p\right)}^{2}}.$$

Since *P_loss_* < 1, it can be devised that
(7)$$E\left[l\right]=\frac{P_{loss}}{1-P_{loss}}.$$

From the analysis, it can be concluded that this method allows a trade-off between the additional delay and the *redundancy factor*. The trade-off is illustrated in the next figures: from Figure 11, it can be observed that the *redundancy factor* mainly depends on the ratio *RTT*/period, and the loss probability does not make any significant difference. From Figure 12, it can be concluded that the loss probability and the period are the two factors that determine the additional delay.

A test battery has been conducted using the same testbed of Section 3, with the implementation of Simplemux *blast* flavor running between the two Raspberry Pi 3B+. As before, the two Raspberries are synchronized via NTP before the test, and two capture files are obtained with Wireshark. The two captures are parsed by a Python script, using the identifier of each packet to calculate the incurred delay.

First, Table 6 gives some results obtained in the testbed, using typical values of the RTT: 20, 50, and 100 ms [28]. The RTT and the loss probability (*P_loss_*) are determined by the scenario, so the period is the parameter that can be tuned by the network manager: if a very short value is set, the delay caused by packet loss can be kept into very low values (in the order of the period plus 0.05 to 0.22 ms).

As a counterpart, the redundancy can scale up to a ×4, ×5, or even a ×10 factor. This could potentially lead to traffic congestion if not managed appropriately. Besides maintaining the period at an optimal value, another strategy to keep redundancy at acceptable levels involves transmitting only the most critical packets (e.g., the trips) via Simplemux *blast*, while the rest are sent without confirmation. VLAN tags can be effectively utilized to categorize the packets.

The value of the period will therefore be limited by the redundancy allowed by the available bandwidth. It is clear that the method can be beneficial for loss-prone networks with high RTT: as an example, a copy of the packet would be available 22.22 ms later instead of waiting for the RTT (100 ms, see the last row of Table 6).

Considering that this method always delivers all the frames, the important performance indicator is not the loss rate but the additional delay caused by packet loss, with different burstiness levels. We will first present two detailed examples, and some averaged results will then be reported.

Figure 13 and Figure 14 show two sets of 40 faults, each of them generating a burst of GOOSE frames jointly with periodic ones. The period is set to 10 ms. In the first case (Figure 13), with a low RTT, a low loss rate (1%), and no bursty losses, the additional delay is kept very low: an average of 0.36 ms, up to 10 ms in some few cases (and 16 ms in one case). If compared with TCP (see Figure 8, obtained in the very same network conditions), the advantage in terms of delay is clear: in this case, the maximum delay is 16 ms, whereas with TCP, it was up to 455 ms. The processing delay in the Raspberry is roughly 0.1 ms. In a real deployment, this delay could even be reduced by using more specific hardware.

Things become more complicated in Figure 14. Since the loss rate is 10%, packets are lost in bursts (ABEL = 10), and the RTT is higher. In this case, the maximum delay becomes 330 ms, although it is 3.68 ms on average.

The averaged results considering no bursty losses (ABEL = 1) show that the average added delay is usually under 0.5 ms (Table 7). Furthermore, the maximum delay added to a packet was 16.16 ms. The standard deviation remains low.

As reported in Table 8, the effect of bursty losses (ABEL = 10) is noticeable, especially when combined with a high loss rate (10%). In these cases, the variance of the delay grows significantly, with some packets sent more than 30 times (period of 10 ms and delay above 300 ms). However, the average added delay only grows up to 2–3 ms. This can be an interesting improvement, considering that GOOSE frames are sent in bursts, so it is easy for at least one of them to arrive on time.

All in all, the results illustrate the trade-off between the reduction in the added delay and the bandwidth increase. It will be the decision of the network operator to tune the period so the delay is kept to the required limits, always considering the bandwidth limitations imposed by the connection technology and the costs.

In general, it is clear that a profound understanding of the underlying network is essential to make an informed decision between a method without confirmation (such as *R*-GOOSE or VX-GOOSE) and the Simplemux *blast* approach, which continues to send the frame until it is received. If the network exhibits bursty packet loss behavior, it would be more advantageous to implement the latter method, bearing in mind the critical importance of maintaining a stable electrical grid.

## 5. Conclusions

Two proposals for sending tunneled GOOSE frames in a WAMPAC system have been presented and evaluated, and the obtained results illustrate their usefulness. The proposed methods can be convenient for some use cases in which an unreliable network is used for the communications of a WAMPAC system. The ability to decouple communication from security allows an easier integration of the latest security protocols.

Both proposals can be seen as examples of the convergence between IT (Information Technology) and OT (Operational Technology) in the smart grid: VXLAN is a mature IT technology widely used in other fields, published by the IETF, and natively implemented in Linux. Although it was conceived for a very different context (data centers), it can also provide significant advantages in substation automation.

The use of a VXLAN tunnel for sending GOOSE frames (i.e., VX-GOOSE) has two advantages: it has the benefits of *R*-GOOSE, as all the functionality of GOOSE is maintained, but without the need to implement all the specific features of *R*-GOOSE. And the tunnel can make use of a VPN, which may already exist to secure the connection between remote locations, so security can be decoupled from the transmission of information.

VX-GOOSE offers two distinct advantages. Like *R*-GOOSE, it retains all the functionalities of GOOSE but without the necessity to incorporate all its specific features. Additionally, the tunnel can leverage a VPN, which might already be in place, to secure connections between remote locations. This allows for the separation of security measures from the transmission of information. The tests have demonstrated that under normal network conditions, where packet loss does not occur in bursts, only minimal delays are observed. Even with a 10% loss rate, the average delay is a mere 0.6 ms. However, under severe network conditions characterized by bursty loss, the transmission of GOOSE frames over UDP may fail. In some instances, an entire burst of GOOSEs could potentially be lost.

Simplemux *blast* flavor, although not a standard, is a way to ensure the fast delivery of all the frames. The tests have shown that there is a tradeoff between the delay and the redundancy factor. This tradeoff is governed by the main parameter: the period in which frame copies are dispatched. By selecting an optimal value, the delay can be significantly minimized, potentially to just a few tens of milliseconds. The increased bandwidth resulting from redundancy can be mitigated by applying the method only to pertinent packets. This minor drawback is negligible when considering the stakes: maintaining grid stability and safeguarding valuable equipment.

The sending of GOOSE constitutes a relevant use case for Simplemux *blast* flavor, but it can also be useful in other fields where flows require very low delay and delivery guarantees. If that is the case, its standardization could be of interest in the near future.

## Figures and Tables

**Figure 1 sensors-23-08879-f001:**
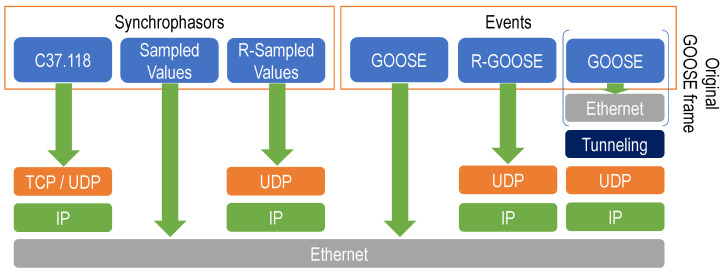
Protocol stack of different communication mechanisms used for substation automation and tunneling proposals (right column).

**Figure 2 sensors-23-08879-f002:**
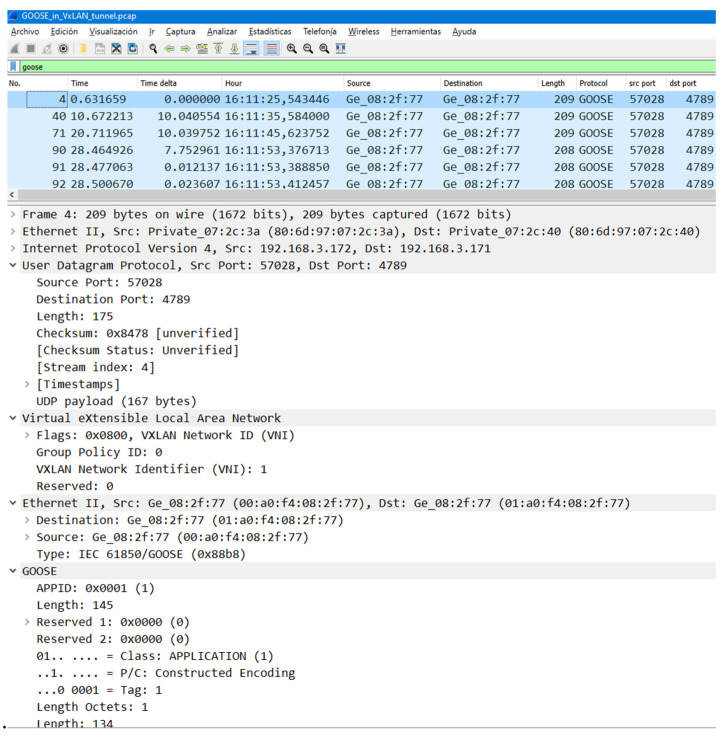
Wireshark capture of VX-GOOSE.

**Figure 3 sensors-23-08879-f003:**
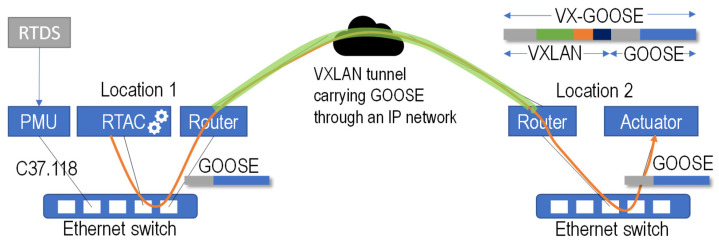
Communication and test setup scheme.

**Figure 4 sensors-23-08879-f004:**
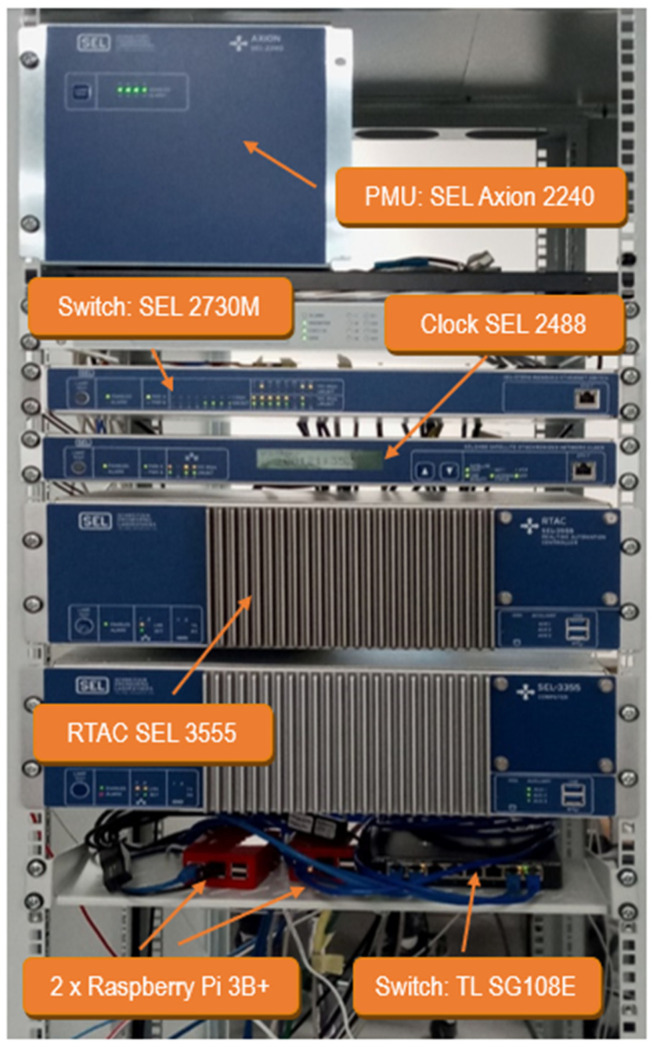
Laboratory testbed (RTDS not shown).

**Figure 5 sensors-23-08879-f005:**
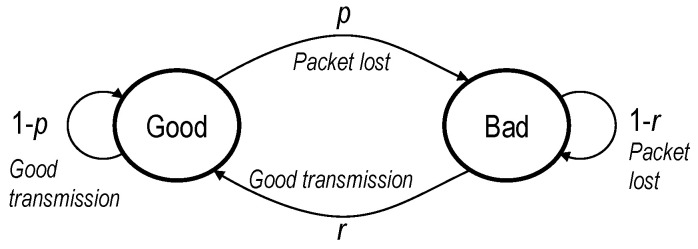
Parameters of the Simple Gilbert Model.

**Figure 6 sensors-23-08879-f006:**
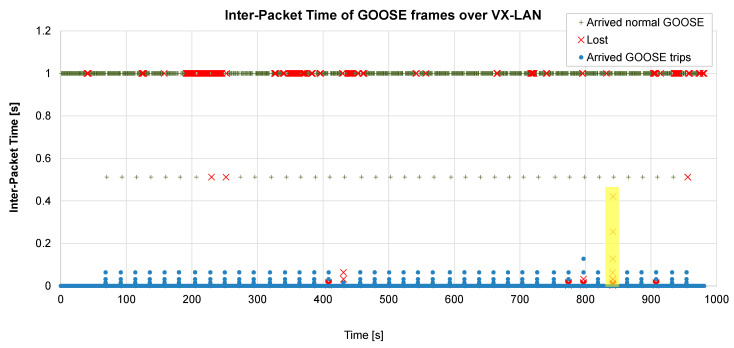
Battery of 40 VX-GOOSE tests. Lost rate 10%, ABEL = 5 frames.

**Figure 7 sensors-23-08879-f007:**
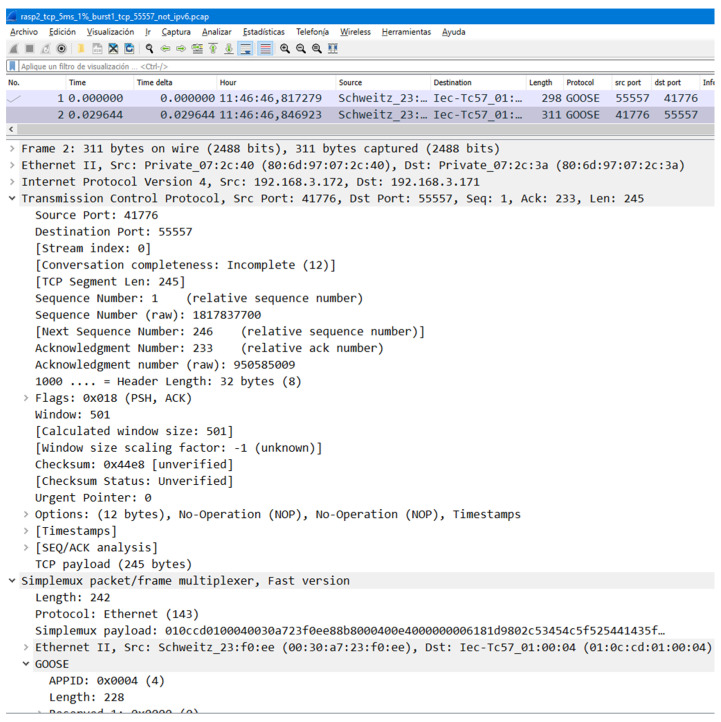
Wireshark capture of GOOSE over Simplemux over TCP.

**Figure 8 sensors-23-08879-f008:**
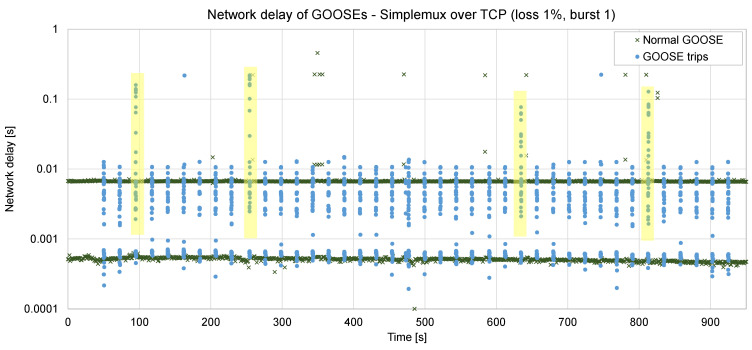
Forty trips sent via Simplemux over TCP, loss rate 1%, ABEL = 1, RTT = 5 ms.

**Figure 9 sensors-23-08879-f009:**
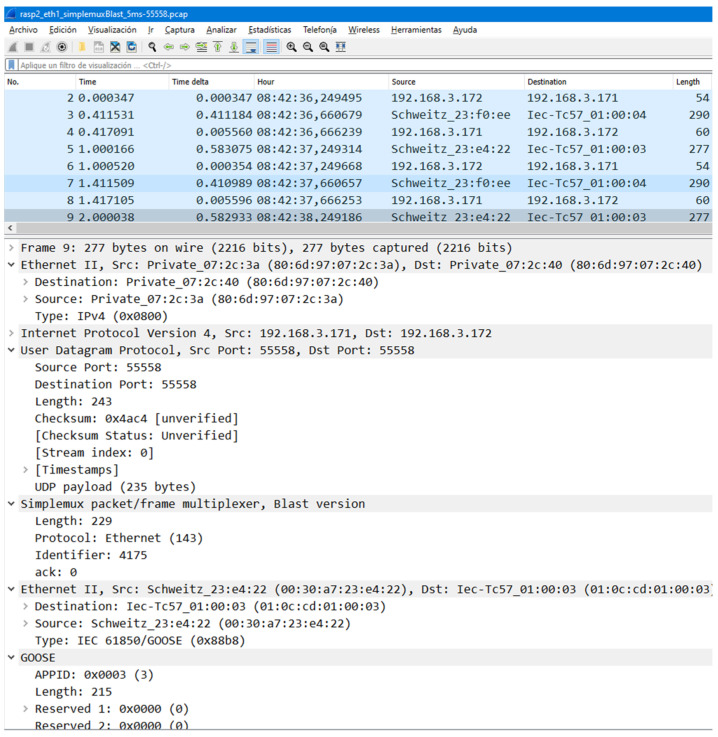
Wireshark capture of Simplemux, *blast* flavor.

**Figure 10 sensors-23-08879-f010:**
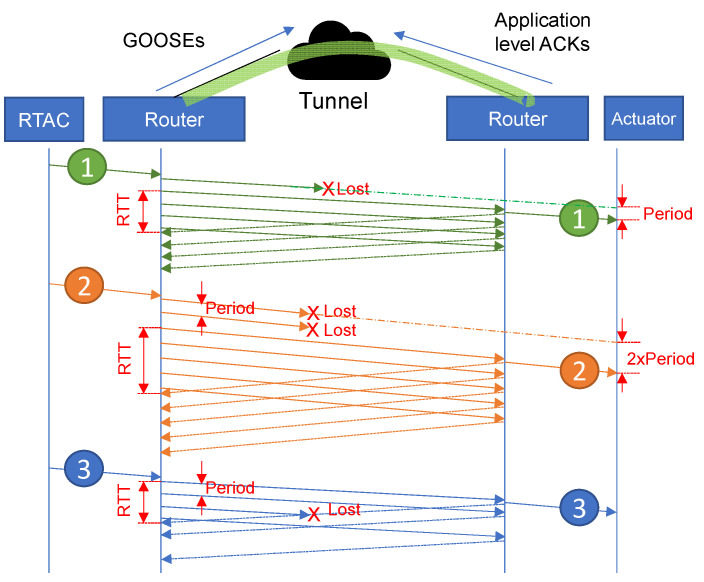
Behavior of Simplemux, *blast* flavor.

**Figure 11 sensors-23-08879-f011:**
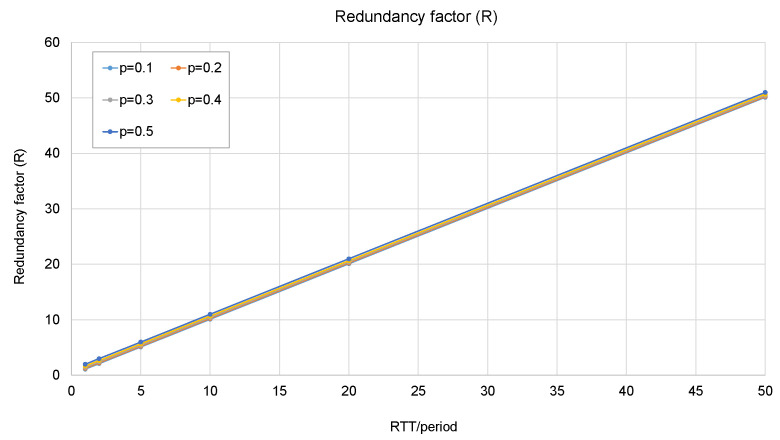
Redundancy factor as a function of *RTT*/period and loss probability.

**Figure 12 sensors-23-08879-f012:**
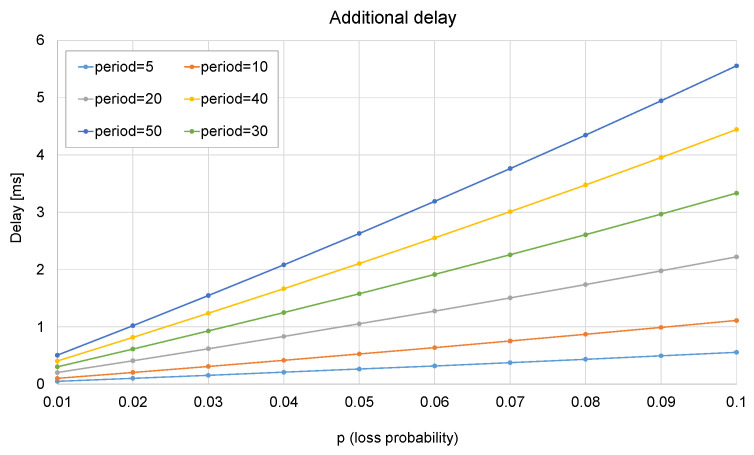
Additional delay as a function of loss probability and the period.

**Figure 13 sensors-23-08879-f013:**
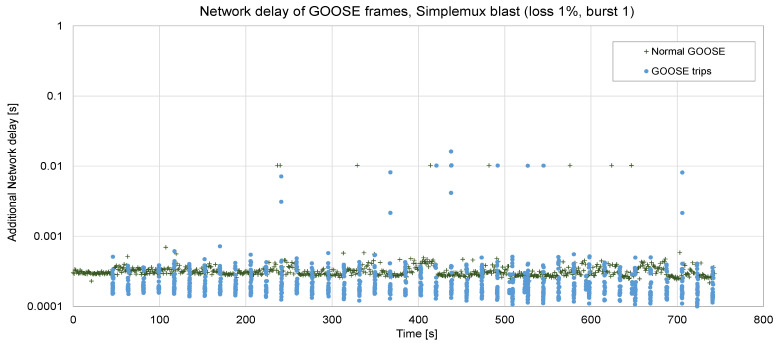
Forty trips sent via Simplemux, *blast* flavor. P = 10 ms, loss rate 1%, ABEL = 1, RTT = 5 ms.

**Figure 14 sensors-23-08879-f014:**
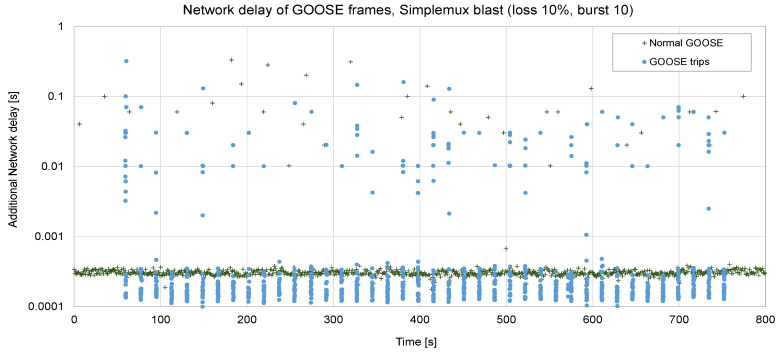
Forty trips sent via Simplemux, *blast* flavor. P = 10 ms, loss rate 10%, ABEL = 10, RTT = 50 ms.

**Table 1 sensors-23-08879-t001:** Protocols for substation automation: summary of the related work.

Ref.	Type	Summary
[4]	Standard	IEC 61850-90-5: *Routed* version of *Sampled Values* and GOOSE protocols
[5]	Standard	IEC 61850-90-1: Exchange of IEC 61850 messages between substations
[7]	Standard	IEEE C37.118: Transmission of synchrophasor data
[8]	Report	Comparing the existing phasor communications protocols
[9]	Standard	Standard: IEC 61850-9-2: Sampled Values: Transmission of samples of the signal inside frames
[10]	Standard	IEC 61869-9: Requirements for digital communications of instrument transformer measurements
[11]	Standard	IEC 61850-8-1: GOOSE: Transmission of event notifications
[12]	Paper	Analysis of teleprotection schemes based on IEC 61850-90-1
[13]	Paper	Sending of GOOSE frames inside GRE datagrams
[14]	Standard	GRE: Generic Routing Encapsulation, a tunneling protocol
[15]	Paper	Sending of GOOSE frames via L2TPv3
[16]	Standard	L2TPv3: Layer Two Tunneling Protocol—Version 3
[17]	Paper	Paper that interprets and implements IEC 61850-90-5 Routed versions of Sampled Values and GOOSE

**Table 2 sensors-23-08879-t002:** Security for substation automation protocols: summary of the related work.

Ref.	Type	Summary
[18]	Standard	IEC/TS 62351: Definition of the security of IEC TC 57 series of protocols
[19]	Paper	Architecture to secure GOOSE and Sampled Values protocols
[20]	Paper	Survey about security assessment and evaluation of VPNs
[21]	Paper	Analysis of WireGuard and other VPN solutions
[22]	Report	Definition of OpenVPN
[23]	Standard	Internet Protocol security, IPsec
[24]	Paper	Analysis and comparison of popular VPN solutions
[25]	Paper	Performance comparison of VPN protocols at the Network layer

**Table 3 sensors-23-08879-t003:** Latency in substation automation networks: summary of the related work.

Ref.	Type	Summary
[26]	Report	Classification of the different components of network delay
[27]	Report	Definition of the components of the delay in large electric systems
[28]	Report	Survey of the different network delays observed in different connections worldwide

**Table 4 sensors-23-08879-t004:** Employed variables.

Variable	Meaning
*p*	Transition probability to the *bad* state
*r*	Transition probability to the *good* state
*P_loss_*	Packet loss probability
ABEL	Average Burst Error Length, i.e., number of packets lost in a row
period	The interval between the sending of two copies of the same Simplemux packet
*l*	Number of packets lost at the beginning of a burst
*R*	Redundancy factor, i.e., the average number of times that a frame is sent via Simplemux
*k*	Number of packets in a row that are lost

**Table 5 sensors-23-08879-t005:** Effect of packet loss on VX-GOOSE (total 40 tests).

Loss Rate	ABEL [Packets]	Num. Delayed Trips	Avg. Delay [ms]	Num. Lost Trips
1%	1	1	0.09	0
5%	1	0	0	0
10%	1	6	0.6	0
1%	5	0	0	0
5%	5	1	0.09	0
10%	5	4	9.19	1
1%	10	1	24.9	0
5%	10	1	3.1	0
10%	10	2	23.8	3

**Table 6 sensors-23-08879-t006:** Examples of *R* and the additional delay.

RTT (ms)	*P_loss_*	E[*l*]	Period (ms)	*R*	Delay Caused by Packet Loss (ms)
20	0.01	0.01	5	4.01	5.05
		0.01	10	2.01	10.101
	0.1	0.11	5	4.11	5.55
		0.11	10	2.11	11.11
50	0.01	0.01	10	5.01	10.101
		0.01	20	3.01	20.202
	0.1	0.11	10	5.11	11.11
		0.11	20	3.11	22.22
100	0.01	0.01	10	10.01	10.101
		0.01	20	5.01	20.202
	0.1	0.11	10	10.11	11.11
		0.11	20	5.11	22.22

**Table 7 sensors-23-08879-t007:** Effect of period and RTT (ABEL = 1).

RTT (ms)	Period (ms)	*P_loss_*	ABEL	Avg Added Delay (ms)	Delay Stdev (ms)	Max Added Delay (ms)
5	5	1%	1	0.32	0.48	5.26
	10	1%	1	0.36	1.01	16.16
	15	1%	1	0.45	1.63	15.24
50	5	1%	1	0.35	0.51	5.41
	10	1%	1	0.36	0.96	10.25
	15	1%	1	0.36	1.27	15.37

**Table 8 sensors-23-08879-t008:** Effect of period and RTT (ABEL = 10).

RTT (ms)	Period (ms)	*P_loss_*	ABEL	Avg Added Delay (ms)	Delay Stdev (ms)	Max Added Delay (ms)
5	10	1%	1	0.36	1.01	16.16
	10	1%	10	0.38	2.71	70.33
	10	10%	10	2.37	17.38	390.30
50	10	1%	1	0.36	0.96	10.25
	10	1%	10	0.46	3.801	120.27
	10	10%	10	3.68	20.209	330.32

## Data Availability

No new data were created or analyzed in this study. Data sharing is not applicable to this article.

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
