# Peer review of "Fast and Reliable Sending of Generic Object Oriented Substation Event Frames between Remote Locations over Loss-Prone Networks"

_sensors, 2023, doi:10.3390/s23218879_

Round 1

Reviewer 1 Report

Comments and Suggestions for Authors

This work explored two  solutions for using tunnels to send GOOSE messages that can fulfill the presented needs in Wide Area Monitoring Protection and Control systems. The approach presented a combination of standard protocols that Virtual Extensible LAN to send tunneled GOOSE frames. Moreover,  Simplemux blast approach based on sending redundant frames to keep the end to end delay below 80 ms. 

The work in total is good, as it tries to present the importance of GOOSE messages in Wide Area Monitoring Protection and Control systems. 

Some suggestions for the authors are shown below:

It is good to clearly present the added value of the work and focus on why this work is important and justifications why the research is needed. 

It is good to briefly define and introduce some standards such as IEC/TS 62351and C37.118

The study doesn't cover the effect of the GOOSE message size on the end to end delay. It is good to look at whether there will be a specific threshold for the GOOSE message size which should be considered in order to meet the end to end latency target. 

The paper doesn't look at it whether the encryption can be used in the R-GOOSE, taking into consideration that the IEC 62351 standard  recommends the use of RSA encryption for message Authentication. 

The discussion section can be restructured in a better way to show the importance of the work. For example I can't see the importance of  Figures 7 & 9 (i.e., Wireshark capture of GOOSE), if there is any valuable findings from that, please highlight that.

It is not clear how the authors got the end to end delay, it is good to incldue that in the paper. 

I suggest adding some recommendations based on the the key findings and share any lessons learned from the test. 

The conclusion section should be extended to cover the key findings and show the key messages from such good work. 

Comments on the Quality of English Language

The English is okay and understandable but it  can be improved 

Reviewer 2 Report

Comments and Suggestions for Authors

  1. The paper's introduction lacks strength and fails to offer substantial information pertaining to the problem addressed in the research.
  2. The author should consider revising the abstract of the paper to provide a more informative overview of its content.
  3. It is imperative that the paper clearly defines the novelty and contributions of the research.
  4. Abbreviations and notations should be appropriately defined upon their initial occurrence and consistently employed throughout the paper.
Comments on the Quality of English Language

N/A

Reviewer 3 Report

Comments and Suggestions for Authors

This paper presents two methods for sending tunneled GOOSE frames in a WAMPAC system that have been presented and evaluated, and the obtained results illustrate their usefulness. They can be convenient for some use cases in which an unreliable network is used for the communications of a WAMPAC system. The ability to decouple communication from security allows for easier integration of the latest security protocols.  The overall presentation of the paper is not enough to accept the present version. The authors should consider the following commands seriously in order to improve the quality of the paper:

1.      For easier comprehension, the authors must incorporate graphical abstracts in the introduction section.

2.      Rather than merely presenting various works in detail, Section 2 should be reorganized using a table that summarizes groups of topics and comments on the main findings.

3.      How does the suggested system handle network traffic and congestion during packet transmission if the packets are delivered repeatedly to prevent packet loss?

4.      To lessen plagiarism in the manuscript, modify the statement on page 2, lines 54-58, in accordance with the requirements, and add the necessary citation. Similarly, for the other areas.

5.      Please supply a table and a definition of the variables so that the acronym and variables can be understood better.

6.      Due to the encapsulation or decapsulation of GOOSE messages, the proposed method imposes any additional network latency; this latency may affect? How real-time GOOSE messages work in substation automation. If so, could you explain?

7.      How does the suggested approach stop unauthorized access? The proposed system is capable of handling it. Please clarify?

8.      What are the statistical reliability and limitations of the proposed work?

Comments on the Quality of English Language

-

Reviewer 4 Report

Comments and Suggestions for Authors

The article analyzes the possibility of achieving fast and reliable sending of GOOSE frames between remote locations (WAMPAC) over loss-prone networks. Two innovative proposals were presented, called VX-GOOSE and Simplemux blast flavor. Based on appropriate analysis confirmed by test results, appropriate practical conclusions were formulated stating that the presented solutions may be useful in various applications where flows require very low delay delivery quarantees The article is written in a clear and readable way, but not least for clarity explanations should be given: All abbreviations such as TCP, GRE, TLS IETF etc should be explained first, Fig. 2 and 6-9 should be enlarged accordingly row (5) what is k? Table2 should be P loss and not p loss.

Round 2

Reviewer 3 Report

Comments and Suggestions for Authors

The quality of this manuscript is now good enough to justify publishing consideration.